# Sacubitril/Valsartan Combination Partially Improves Cardiac Systolic, but Not Diastolic, Function through β-AR Responsiveness in a Rat Model of Type 2 Diabetes

**DOI:** 10.3390/ijms251910617

**Published:** 2024-10-02

**Authors:** Betul R. Erdogan, Zeynep E. Yesilyurt-Dirican, Irem Karaomerlioglu, Ayhanim Elif Muderrisoglu, Kadir Sevim, Martin C. Michel, Ebru Arioglu-Inan

**Affiliations:** 1Department of Pharmacology, Faculty of Pharmacy, Ankara University, Ankara 06560, Türkiye; betul.r.erdogan@gmail.com (B.R.E.); zeynepelifyesilyurt@gazi.edu.tr (Z.E.Y.-D.); irem.karaomerlioglu@gmail.com (I.K.); elifmuderrisoglu@gmail.com (A.E.M.); 2Department of Pharmacology, Faculty of Pharmacy, Gazi University, Ankara 06330, Türkiye; 3Department of Medical Pharmacology, Istanbul Medipol University, Istanbul 34815, Türkiye; 4Department of Internal Medicine, Faculty of Veterinary Medicine, Ankara University, Ankara 06110, Türkiye; ksevim@ankara.edu.tr; 5Department of Pharmacology, Johannes Gutenberg University, 55131 Mainz, Germany

**Keywords:** β-adrenoceptor, diabetes, heart, sacubitril, valsartan

## Abstract

Cardiovascular complications are the major cause of diabetes mellitus-related morbidity and mortality. Increased renin–angiotensin–aldosterone system activity and decreased β-adrenergic receptor (β-AR) responsiveness contribute to diabetic cardiac dysfunction. We evaluated the effect of sacubitril/valsartan (neprilysin inhibitor plus angiotensin receptor antagonist combination) and valsartan treatments on the diabetic cardiac function through β-AR responsiveness and on protein expression of diastolic components. Six-week-old male Sprague Dawley rats were divided into control, diabetic, sacubitril/valsartan (68 mg/kg)-, and valsartan-treated (31 mg/kg) diabetic groups. Diabetes was induced by a high-fat diet plus low-dose streptozotocin (30 mg/kg, intraperitoneal). After 10 weeks of diabetes, rats were treated for 4 weeks. Systolic/diastolic function was assessed by in vivo echocardiography and pressure–volume loop analysis. β-AR-mediated responsiveness was assessed by in vitro papillary muscle and Langendorff heart experiments. Protein expression of sarcoplasmic reticulum calcium ATPase2a, phospholamban, and phosphorylated phospholamban was determined by Western blot. Sacubitril/valsartan improved ejection fraction and fractional shortening to a similar extent as valsartan alone. None of the treatments affected in vivo diastolic parameters or the expression of related proteins. β_1_-/β_2_-AR-mediated responsiveness was partially restored in treated animals. β_3_-AR-mediated cardiac relaxation (an indicator of diastolic function) responses were comparable among groups. The beneficial effect of sacubitril/valsartan on systolic function may be attributed to improved β_1_-/β_2_-AR responsiveness.

## 1. Introduction

Sacubitril/valsartan, a first-in-class neprilysin inhibitor plus angiotensin receptor antagonist combination (ARNI), was approved in 2015 for the treatment of heart failure with reduced ejection fraction (HFrEF) [1]. Sacubitril, a neprilysin inhibitor, increases natriuretic peptide (NP) levels, which has a protective effect on the heart [2]. Valsartan, an angiotensin II (Ang II) type 1 receptor blocker (ARB), prevents the long-term adverse effects of renin–angiotensin–aldosterone system (RAAS) activation on cardiac function [3]. Treatment with sacubitril/valsartan has been shown to be superior to enalapril in reducing heart failure-related hospitalization and cardiovascular and all-cause mortality in patients with HFrEF [4]. A meta-analysis showed that sacubitril/valsartan reversed cardiac remodeling in patients with HFrEF compared with ARBs or angiotensin-converting enzyme inhibitors (ACEIs) alone [5]. There are ongoing clinical studies investigating the efficacy, safety, and tolerability of the sacubitril/valsartan combination in heart failure and other diseases, including diabetes mellitus (hereafter referred to as diabetes) [6].

Diabetes is a chronic, endocrine disease that affects hundreds of millions of people worldwide. People with diabetes are two to three times more likely to develop cardiovascular disease than non-diabetics [7]. Cardiovascular complications include a major risk factor for mortality in individuals with both type 1 and type 2 diabetes [8]. Diabetic cardiomyopathy (DCM) is a dysfunction of the cardiac muscle that occurs independently of hypertension or coronary artery disease in diabetes. DCM can lead to heart failure in the late stages of the disease [9]. Hyperglycemia, insulin resistance, and hyper/hypoinsulinemia in diabetes trigger many mechanisms, including the accumulation of reactive oxygen radicals and advanced glycation end products, RAAS activation, microvascular and endothelial dysfunction [10], cardiac stress, hypertension, atherosclerosis [8], vascular stiffening, and circulatory dysfunction [11], leading to diabetic cardiomyopathy. Ang II also contributes to the development and progression of diabetic cardiomyopathy [10] by increasing reactive oxygen species and/or causing hypertrophy and fibrosis directly through the Ang II type 1 receptor [12].

Decreased levels of NPs are associated with obesity [13], insulin resistance [14], and the development of diabetes [15]. Therefore, it was proposed that therapeutic agents that increase NPs may be beneficial in the treatment of diabetes and its complications [16,17]. Neprilysin inhibitors have been suggested to have glycemic control effects by increasing glucagon-like peptide-1, vasoactive intestinal polypeptide, and insulin B-chain, independently of their effect on NPs [18]. Nevertheless, neprilysin inhibition may lead to increased levels of human islet amyloid polypeptide, which may have a deleterious effect on pancreatic cells [19]. Therefore, its combination with ARBs is suggested as a promising therapeutic approach in type 2 diabetes to prevent the possible deleterious effect of neprilysin inhibition alone [20].

Alterations in β-adrenergic receptor (β-AR) signaling contribute to the cardiac dysfunction in diabetes [21,22]. Sympathetic overdrive in diabetes [23] leads to blunted β_1_- and β_2_-AR-mediated cardiac contractile responses [24,25], which may occur in part due to reduced expression of β_1_- and β_2_-ARs [26,27]. On the other hand, β_3_-AR-mediated cardiac relaxation (an indicator of diastolic function) [28,29] as well as receptor expression are increased in diabetes [24,26,27]. Elevated β_3_-AR expression plays a protective role against the deleterious effects of catecholamines in cardiac tissue [30]. Similarly, sympathetic overdrive results in reduced β_1_- and β_2_-AR-mediated responsiveness in heart failure [31]. Impaired β-AR signaling in pathological conditions can affect the function of several downstream components, such as sarcoplasmic reticulum calcium ATPase 2a (SERCA2a), phospholamban (PLN), and phosphorylated phospholamban (p-PLN).

Drugs that target the RAAS (ARBs and ACEIs) [32] and SGLT2 inhibitors [33] are recommended for patients with HF and diabetes, as well as for patients with HF and especially type 2 diabetes, respectively. Considering the role of RAAS and NPs in diabetes and the beneficial effects of these systems on the heart, the sacubitril/valsartan combination may also be a promising therapeutic approach for diabetic cardiac dysfunction. The sacubitril/valsartan combination improves glucose homeostasis in patients with concomitant HFrEF and diabetes [34]. Moreover, sacubitril/valsartan has been shown to reduce sympathetic nervous system activity in patients with HFrEF [35]. There are few preclinical studies investigating the effect of sacubitril/valsartan on the diabetic heart. Based on the results of these studies, reduced fibrosis [36,37], inflammation and apoptosis [38,39], oxidative stress [38], and endoplasmic reticulum stress [39] have been suggested as possible mechanisms underlying this effect. Although there is no direct link between sacubitril/valsartan and cardiac β-ARs, the question arises whether a combination of sacubitril/valsartan affects β-AR-mediated responses in the diabetic heart, which are impaired in many diabetes models.

Based on these findings, the current study was conducted to investigate (1) the effect of sacubitril/valsartan on cardiac contraction/relaxation in comparison with valsartan using β-AR agonists and (2) the effect of these treatments on diastolic downstream components in a rat model of type 2 diabetes, induced by a high-fat diet (HFD) plus low-dose streptozotocin (STZ).

## 2. Results

### 2.1. General Characteristics of Animals

The body weight of diabetic rats at the end of this study was reduced compared to control rats, as typically observed in this model. Neither sacubitril/valsartan nor valsartan treatment reversed the loss of body weight in diabetes (Figure 1a). Blood glucose levels at the end of the study were higher in diabetic rats, as expected. However, blood glucose levels in rats treated with either sacubitril/valsartan or valsartan were markedly higher than in diabetic rats (Figure 1b). In addition, sacubitril/valsartan caused a greater increase in blood glucose levels than valsartan throughout the treatment period (Appendix A). Heart weight was comparable between groups (Figure 1c). The ratio of heart weight to body weight (HW/BW), an indicator of cardiac hypertrophy, was higher in diabetic rats compared to the control group. This ratio was not improved by either treatment (Figure 1d). Blood glucose levels of diabetic and treated diabetic rats were higher than those of control rats at each time point of the oral glucose tolerance test (OGTT) experiment (Appendix A and Appendix A). There was no difference in fasting insulin levels between the experimental groups, but the Homeostatic Model Assessment-Insulin Resistance (HOMA-IR) index was increased in diabetic rats and was not attenuated in treated diabetic rats (Appendix A). Plasma cholesterol and triglyceride levels were substantially increased in diabetic rats compared with the control group. Neither treatment reversed the increase in plasma cholesterol and triglyceride levels (Appendix A).

### 2.2. In Vivo Cardiac Parameters

#### 2.2.1. Pressure–Volume (PV) Loop Analysis

Basal blood pressure level was measured in rats under anesthesia using a pressure–volume catheter placed in the carotid artery. Systolic blood pressure, diastolic blood pressure, and mean arterial pressure were comparable between the groups. Heart rate (HR) was decreased in the diabetic group and was not normalized with either treatment approach. End systolic and end diastolic volumes (ESV and EDV) were slightly increased in the diabetic group and not normalized by treatment. The indices of these parameters were significantly higher in diabetic rats and not reversed by treatment. While stroke volume (SV) was comparable between the groups, surprisingly, the SV index was markedly increased only in diabetic rats compared to the control group (Table 1).

End systolic pressure (ESP), which indicates the systolic function of the heart, decreased in the diabetic group, and both treatments had no effect on this parameter (Figure 2a). The rate of contraction, which is another indicator of systolic cardiac function, was significantly reduced in the diabetic group and was not affected by the treatments (Figure 2b). 

End diastolic pressure (EDP), one of the diastolic function parameters of the heart, was similar between the groups (Figure 3a). Another indicator of cardiac diastolic function, the rate of relaxation (dP/dt_min_), was decreased in diabetic rats compared to control, and both treatment approaches did not normalize this parameter (Figure 3b). Tau Weiss and Tau Glantz values, the time constants for isovolumic relaxation, were significantly increased in diabetic rats, and neither sacubitril/valsartan nor valsartan treatment ameliorated Tau values (Figure 3c and Figure 3d, respectively).

The preload independent cardiac parameters: the slope of end systolic pressure–volume relationship (ESPVR), end diastolic pressure–volume relationship (EDPVR), and preload recruitable stroke work (PRSW) were comparable between groups (Table 2).

#### 2.2.2. In Vivo Echocardiography Analysis

In vivo echocardiographic analysis showed that ejection fraction (EF) (%) and fractional shortening (FS) (%) were markedly reduced in diabetic rats. This indicates impaired systolic function due to diabetes. These parameters were improved by both sacubitril/valsartan and valsartan treatment to a similar level (Figure 4 and Table 3). Interventricular septum thickness at end diastole index (IVSId), left ventricular internal dimension at end diastole index (LVIDId), left ventricular internal dimension at end systole (LVIDs), left ventricular internal dimension at end systole index (LVIDIs), and left ventricular posterior wall thickness at end systole index (LVPWIs) values were significantly higher in the diabetic group. Both treatments improved IVSId, and only valsartan improved LVIDs values. Cardiac output (CO) was markedly reduced in the diabetic group and was not reversed by treatments. Other echocardiographic parameters were comparable between groups (Table 3).

### 2.3. β-AR-Mediated Responsiveness

In the papillary muscle preparation, the contractile response induced by isoprenaline, a non-selective β-AR agonist, was reduced in the diabetic group, as expected. Both sacubitril/valsartan and valsartan treatment slightly improved the contractile response compared with the diabetic group to a similar level (Figure 5a,b). Despite a noticeable effect of the treatment approaches on isoprenaline-induced contraction, the effect was not statistically significant between groups due to a high standard deviation. In the papillary muscle, forskolin-mediated contractile response was partially higher in sacubitril/valsartan-treated diabetic rats compared to other groups (Figure 5c).

In the Langendorff heart preparation, the % change in left ventricular developed pressure (LVDP) induced by CL 316,243, a selective β_3_-AR agonist, did not differ between groups (Figure 6a). Moreover, the rate of contraction (dP/dt_max_) and relaxation (% basal value) (Figure 6c and Figure 6d, respectively) and the relaxation response induced by 10^−8^ M CL 316,243 (considered as E_max_ value) (Figure 6b) were comparable in all groups.

### 2.4. Protein Expression of Diastolic Components

SERCA2a expression was reduced in the diabetic group and was not reversed by either treatment approach (Figure 7a). The p-PLN/PLN ratio, which indicates a reduced inhibitory activity of PLN on SERCA2a function, was markedly reduced in diabetic rats and was not normalized with treatments (Figure 7b). In addition, the PLN/SERCA2a ratio was comparable between groups (Figure 7c).

## 3. Discussion

To the best of our knowledge, the present study is the first to investigate the effect of the sacubitril/valsartan combination on cardiac contraction and relaxation using β-AR agonists and on the expression of proteins related to diastolic function. In this study, we aimed to investigate the effect of sacubitril/valsartan on β-AR responsiveness. Our results can be summarized as follows:

Sacubitril/valsartan and valsartan were comparable and slightly increased the isoprenaline-induced contractile responses mediated by β-AR, which were impaired in diabetic rats.

In vivo diastolic parameters and the protein expression of the diastolic components were not affected by the sacubitril/valsartan and valsartan treatments.

### 3.1. Critique of the Study Design and Experimental Model

The type 2 diabetes model induced by HFD plus a low dose of STZ was chosen for this study because this experimental model accurately reflects hyperglycemia, hyperlipidemia, and insulin resistance observed in type 2 diabetic patients [40,41]. Furthermore, the HFD plus low-dose STZ diabetes model resembles the late stage of type 2 diabetes [42]. Considering the increasing prevalence of type 2 diabetes worldwide and its cardiovascular complications, the HFD and low-dose STZ-induced diabetes model is a suitable experimental model for the current study. Based on the findings, such as elevated fasting insulin levels, high HOMA-IR index, and increased triglyceride levels, we can conclude that this experimental model has been established. On the other hand, the diabetic groups in this model commonly show weight loss [42,43,44,45], which, clinically, is a common feature of type 1 and not type 2 diabetes. Nevertheless, there are other studies reporting weight loss in this diabetic model.

The randomization for group allocation was performed after the acclimatization period, as prespecified. The blood glucose levels of both sacubitril/valsartan- and valsartan-treated diabetic rats were numerically higher than those of untreated diabetic rats at the end of the study. However, interpretation of these data is hampered by the fact that rats with higher blood glucose levels were randomly assigned to the treated groups. Furthermore, sacubitril/valsartan treatment resulted in a greater increase in blood glucose levels than valsartan treatment alone in the time course analysis (Appendix A). Unlike our findings, sacubitril/valsartan combination improved glucose homeostasis in heart failure patients with concomitant diabetes [34] and in high-fat- and high-fructose-fed rats [46]. In addition, studies have shown that valsartan and other ARBs have little or no effect on blood glucose levels [3]. We found markedly increased cholesterol and triglyceride levels in diabetic rats, and both treatments failed to reduce this increase (Appendix A). However, sacubitril/valsartan has been shown to reduce triglyceride levels in patients with HFpEF [47]. On the other hand, preclinical studies have reported conflicting results on the effect of the sacubitril/valsartan combination on blood glucose and lipid levels. For example, sacubitril/valsartan treatment improved blood glucose and lipid parameters in type 1 diabetic rats [48] and ameliorated blood lipid levels, but not glucose levels, in Zucker obese rats [49]. However, it did not affect either blood glucose or lipid levels in the genetic model of type 2 diabetic mice [50]. The different effects of sacubitril/valsartan may be explained by the different animal models used in these studies.

### 3.2. Cardiac Hypertrophy

Hypertrophy is a common feature of the diabetic heart [51,52] and the HW/BW ratio is an indicator of cardiac hypertrophy. In the current study, the HW/BW ratio was greater in diabetic rats. However, neither sacubitril/valsartan nor valsartan treatments improved this ratio. Consistent with this finding, neither treatment approach had a beneficial effect on increased LVIDId, LVIDIs, and LVPWIs, based on echocardiographic analysis. Sacubitril/valsartan improved only IVSId, and valsartan improved both IVSId and LVIDs.

Improved cardiac hypertrophy with sacubitril/valsartan treatment has been shown in diabetes [38,39] and in other disease models [53,54,55]. However, supporting our data, some studies revealed that the combination did not show an improvement in cardiac hypertrophy. For instance, Miyoshi et al. reported that sacubitril/valsartan attenuated cardiac fibrosis but had no effect on cardiac hypertrophy [56]. Similarly, both treatments failed to attenuate cardiac hypertrophy in Zucker obese rats [57]. Hyperglycemia is one of the major mechanisms leading to cardiac hypertrophy in diabetes [10]. The lack of a beneficial effect of the treatment approaches on cardiac hypertrophy in our study may be due to the lack of an effect of these approaches on the elevated blood glucose levels. In support of this, sacubitril/valsartan treatment reversed cardiac remodeling in non-diabetic heart failure patients but not in diabetic heart failure patients [58]. While most studies suggest a beneficial effect of sacubitril/valsartan on cardiac remodeling, further studies are needed to reach a definitive conclusion in diabetic models.

### 3.3. Cardiac Hemodynamic Parameters

The parameters measured by our techniques are used to define cardiac function in preclinical studies. As there are no formally agreed criteria for systolic and diastolic dysfunction in rodents, the terms systolic and diastolic dysfunction are used to indicate functional impairment for parameters that differ from healthy animals. Diastolic and/or systolic dysfunction are important features of the diabetic heart [59,60]. Diastolic dysfunction is an early symptom of diabetes, whereas systolic dysfunction is observed in the late stage of the pathology [9]. Based on our echocardiographic analysis, EF (%) and FS (%) were markedly reduced in diabetic rats, indicating systolic dysfunction. Furthermore, these parameters were improved with each treatment. Impaired mitochondrial function [61] and increased oxidative stress in the mitochondria are contributing factors in diabetic cardiomyopathy [62]. In this regard, sacubitril/valsartan has been shown to inhibit the production of mitochondrial reactive oxygen species in doxorubicin-treated H9c2 cells [63]. Furthermore, the beneficial effects of sacubitril/valsartan on cardiac function by alleviating mitochondrial function/activity in cardiac pathologies have been demonstrated by others [64,65]. Thus, the improved contractile response in the current study may be related to favorable effects of sacubitril/valsartan on mitochondrial oxidative stress or function. However, this hypothesis needs to be tested in future studies. Twelve months of treatment with sacubitril/valsartan resulted in a greater increase in EF (%) compared with valsartan alone (10.0% vs. 4.6%, respectively) in diabetic patients with acute myocardial infarction [66]. Preclinical studies also support the superiority of sacubitril/valsartan over valsartan alone in improving systolic function in diabetes [36,39] or in the myocardial infarction model [67]. In contrast to these studies, we found that the sacubitril/valsartan combination had a similar effect as valsartan on systolic function. In line with the reduction in EF and FS, dP/dt_max_ was also diminished in the diabetic group, further confirming systolic dysfunction due to diabetes. On the other hand, neither treatment corrected this parameter.

In patients with HFrEF, sacubitril/valsartan treatment ameliorated diastolic and systolic echocardiographic parameters, which is associated with clinical improvement [68]. In addition, switching from ACEi/ARB to sacubitril/valsartan improved both diastolic and systolic cardiac function in patients with HFrEF [69,70]. In the current study, the diastolic dysfunction was also confirmed using the in vivo PV loop analysis by reduced dP/dt_min_ and increased Tau values. Furthermore, increased ESV and EDV indices support the impairment of diastolic function. None of these parameters improved in the treatment groups. Unlike our findings, Tau constant decreased and some hemodynamic parameters improved in rats with myocardial infarction treated with sacubitril/valsartan [71]. Despite the same drug dose and duration of treatment, the discrepancy between the study by von Lueder et al. and ours could be partly explained by the differences in the animal models. In another study, sacubitril/valsartan treatment improved diastolic function parameters (i.e., E’/A’ ratio, diastolic stiffness, and myocardial performance index) better than valsartan alone in Zucker obese rats [57]. However, diastolic function was preserved in rats treated with sacubitril/valsartan but not in rats treated with valsartan in the HFpEF model [72]. Although the doses of the drugs are the same, the differences in results between these studies and ours could be due to the duration of the treatment (10 weeks [57] and 8 weeks [72]) and/or the different animal models. In our study, the parameters of diastolic dysfunction were measured by PV loop analysis, whereas in other studies they were mostly measured by echocardiography. Nonetheless, it seems unlikely that the difference in methodology would have impacted our findings.

### 3.4. β-Adrenergic Responsiveness

β_1_-AR-mediated contractile response to isoprenaline was markedly reduced in diabetic rats. Blunted β_1_-AR-mediated contractility in diabetic rat heart has also been reported in previous studies [73,74]. Impairment of cardiac contractile response in diabetic heart has been attributed to reduced β_1_-AR expression levels [73,75,76] and hyperinsulinemia-induced desensitization of β_2_-AR [77]. Although both treatments improved the decline in contractile performance, the results were not statistically significant due to the high standard deviation. Nevertheless, considering the E_max_ values, it could be assumed that sacubitril/valsartan is more effective than valsartan alone on the contractile response. This effect may be related to the sympatho-inhibitory effect of sacubitril/valsartan [35] and/or improved Ca^2+^ homeostasis [67,78]. Moreover, ryanodine receptors (RyR) play an essential role in cardiac contraction by mediating Ca^2+^ release from the SR. RyR dysfunction seen in diabetic cardiomyopathy or heart failure may be related to sympathetic overdrive and oxidative stress, as impaired RyR function due to hyperadrenergic/oxidative overload is one of the proposed mechanisms leading to Ca^2+^ leak in several pathologies, including heart failure and diabetes [79]. RyR hyperphosphorylation and Ca^2+^ leak due to the hyperadrenergic state (increased norepinephrine release) were seen in the failing canine heart [80]. Valsartan treatment corrected the RyR hyperphosphorylation and the abnormal Ca^2+^ leak by inhibiting norepinephrine release and stimulating its uptake into the synaptic pool. Furthermore, valsartan treatment increased the density of β-ARs and the dobutamine-induced contractile response mediated by β-ARs [80]. Sacubitril/valsartan has also been shown to have a favorable effect on cardiac oxidative stress in a variety of pathological conditions [81,82,83,84]. In this context, improvement of oxidative stress and hyperadrenergic state by sacubitril/valsartan treatment may have a beneficial effect on RyR function and thus on cardiac contractility. Based on this hypothesis, our results showing slightly increased β_1_- and β_2_-AR-mediated contractility in sacubitril/valsartan- and valsartan-treated groups may be related to improved RyR function/phosphorylation due to alleviation of cardiac oxidative stress. However, we were not able to evaluate the function/phosphorylation of the RyR and the oxidative stress of the heart in the present study.

As β_1_- and β_2_-ARs mediate the cardiac contraction through the adenylyl cyclase (AC)-cyclic adenosine monophosphate (cAMP)-cAMP-dependent protein kinase A (PKA) downstream pathway [85], we also assessed forskolin-induced contractility. Despite slightly higher values in the sacubitril/valsartan-treated group, the contractile response was comparable between the groups.

CL 316,243, a β_3_-AR agonist, induced cardiac relaxation in a concentration-dependent manner in the Langendorff perfused heart preparation. Although the cardiac relaxation response was slightly lower in diabetic rats, there was no statistically significant difference between the groups. Interestingly, in previous studies, we found that β_3_-AR-mediated cardiac relaxation was enhanced in the diabetic heart [28,86,87]. Of note, those studies have been performed in an STZ diabetic model that mimics type 1 diabetes. On the other hand, the model we used in the current study mimics type 2 diabetes. Consistent with our findings, Derkach et al. reported that cardiac β_3_-AR-mediated relaxation response is preserved in HFD and low-dose STZ-induced diabetic rats [88]. Another explanation for the different results may be that HFD and low-dose STZ-induced diabetes presents a milder cardiac phenotype, which contributes to preserved cardiac β_3_-AR-mediated relaxation.

### 3.5. Protein Expression of Diastolic Components

The membrane depolarizes and Ca^2+^ enters the cardiomyocytes during systole. The ryanodine receptor phosphorylated by PKA senses the increased amount of Ca^2+^ in the cell, and Ca^2+^-induced Ca^2+^ release from the sarcoplasmic reticulum (SR) occurs, initiating a contractile response in cardiomyocytes [89]. Reuptake of cytosolic Ca^2+^ into the SR by SERCA2a is a critical step in cardiac relaxation and subsequent contraction [89]. Reduced SERCA2a protein density is a common feature in the pathogenesis of diabetic cardiac dysfunction [73,90] and heart failure [91]. The decrease in SERCA2a protein expression may be due to reduced phosphorylation of PLN, which has an inhibitory effect on SERCA2a [92,93].

Changes in SERCA2a expression are associated with diastolic parameters, such as relaxation rate and Tau constant [94]. Cardiac SERCA2a expression was decreased in 6-week-STZ-induced diabetic rats, and impaired diastolic function was attributed to decreased SERCA2a expression [90]. Similar results have been reported by other researchers in 8- and 12-week-STZ-induced diabetic rats [73,95,96]. In line with these studies, we found that SERCA2a was downregulated in diabetic rats, which may have contributed to the impaired relaxation rate and Tau constant values. However, both treatment approaches failed to normalize the reduced SERCA2a protein expression.

PLN regulates SERCA2a activity through its inhibitory effect. The p-PLN to PLN ratio is an indicator of PLN activity. A decreased ratio of p-PLN to PLN indicates inadequate phosphorylation of PLN during diastole, leading to impaired cardiac relaxation. A decrease in the ratio of p-PLN to PLN, due to or independent of adrenergic stimulation, has been reported in type 2 diabetic animals [22]. In our study, the p-PLN to PLN ratio was decreased in the diabetic group, which correlates with impaired in vivo diastolic parameters. Treatment approaches failed to normalize this ratio. Changes in PLN to SERCA2a ratio have been suggested to involve diastolic dysfunction in type 2 diabetic heart [97]. However, we found a similar PLN to SERCA2a ratio in our study.

## 4. Materials and Methods

### 4.1. Animals and the Study Protocol

The study protocol had been approved by the animal welfare committee of Ankara University (permit 2016-23-198) and was in line with NIH Guidelines for Care and Use of Laboratory Animals. Four-week-old male Sprague Dawley rats were obtained from the Department of Molecular Biology and Genetics, Laboratory Animals Unit, Bilkent University and housed under a 12:12 h light/dark cycle with chow and water ad libitum. After a 2-week acclimatization period, the rats were randomized into groups (control (C), diabetic (D), sacubitril/valsartan-treated diabetic (SV), and valsartan-treated diabetic (V)). These rats were given either a HFD (45% kcal fat) or a control diet (10% kcal fat). After 4 weeks, the rats on the HFD received an intraperitoneal injection of STZ (30 mg/kg, in citrate buffer, pH 4.5), while the rats on the control diet received an injection of vehicle (citrate buffer, pH 4.5). One week after STZ injection, the rats with blood glucose levels above 200 mg/dL were considered diabetic. Some rats received a second (37 rats), third (15 rats), or fourth (11 rats) STZ injection, if necessary. The need for multiple doses of STZ injection to establish the diabetes model caused a delay in the experimental protocol. Following 10 weeks of diabetes, group SV was treated with sacubitril/valsartan (68 mg/kg/day, oral gavage), and group V was treated with valsartan (31 mg/kg/day, oral gavage) for 4 weeks. Groups C and D received vehicle treatment by oral gavage. Sacubitril/valsartan and valsartan treatment dosage and treatment period were applied according to “Guidance to investigators for formulating and administering LCZ696-ABA and valsartan to rats” by Novartis. Body weight and plasma glucose levels, quantitative indicators of glycemia, were measured weekly. Because of the delay in diabetes establishment, the rats reached the point for in vitro experiments at 25–29 weeks of age instead of the planned time point (24 weeks of age). Rats were anesthetized under isoflurane (2% infusion), and the hearts were excised. Tissues and plasma samples were stored at −80 °C for further experiments. Plasma cholesterol and triglyceride levels were measured at the time of sacrifice using enzymatic colorimetric kits. Western blot experiments were performed on a new set of animals, not on the tissues of the animals that were used for in vivo and in vitro functional studies, due to breakdown of the −80 °C freezer during the study.

### 4.2. Oral Glucose Tolerance Test (OGTT)

OGTT was performed during the third week of the sacubitril/valsartan and valsartan treatment. The rats were fasted for 12 h, and 2 g/kg glucose was administered by oral gavage. Blood glucose levels were measured at 0, 30, 60, 90, and 120 min using a glucometer via the tail vein. Blood samples were collected in EDTA tubes, and plasma was separated. Plasma samples were stored at −80 °C for insulin level measurements. The HOMA-IR index was calculated as previously described [(fasting blood glucose (mmol/L) × fasting insulin level (µIU/mL))/22.5] [98]. Fasting plasma insulin levels during the OGTT analysis were measured by using ELISA kits.

### 4.3. In Vivo Pressure–Volume (PV) Loop Analysis

Rats were injected with 3000 U/kg of intraperitoneal heparin 10 min before anesthesia, and then anesthetized with isoflurane (2% infusion) at the end of 4-week sacubitril/valsartan and valsartan treatment. Body temperature was maintained at 37 °C. The right carotid artery was dissected, and a small incision was made on the surface of the carotid artery. After the catheter (Millar Instruments, Houston, TX, USA) was inserted into the carotid artery, the blood pressure was recorded. The catheter was then placed into the left ventricle. After a 10-min stabilization period, PV loops were recorded at constant preload. The following hemodynamic parameters were measured and analyzed using by PV loop data analysis system (Transonic SciSense Inc., London, ON, Canada): end systolic pressure (ESP), end diastolic pressure (EDP), end systolic volume (ESV), end diastolic volume (EDV), stroke volume (SV), cardiac output (CO), heart rate (HR), rate of contraction (dP/dt_max_), rate of relaxation (dP/dt_min_), and isovolumic relaxation time constant (Tau). Volume-dependent hemodynamic parameters were calculated as body weight normalized index values to eliminate the effect of body weight changes due to diabetes. Preload independent cardiac parameters (preload recruitable stroke work (PRSW), the slope of end systolic pressure–volume relationship (ESPVR), and end diastolic pressure–volume relationship (EDPVR)) were then measured and analyzed by occlusion of the inferior vena cava for 5 s.

### 4.4. In Vivo Echocardiography Experiment

Due to logistical reasons, the echocardiography experiment was carried out 2 to 3 weeks after the start of the sacubitril/valsartan and valsartan treatment period. The rats were anesthetized using 2% isoflurane inhalation, placed in the left lateral recumbent position, and scanned using the Arietta V60 ultrasound cardiovascular system echo scanner (Hitachi Healthcare Inc., Tokyo, Japan) with a pediatric S31 (2–9MHz) phased array transducer and a high temporal and spatial resolution cardiac application. The transmission frequency was 10 MHz; the depth was 2.5 cm; and the frame rate was 225 frames per second. The conventional two-dimensional (2D) short-axis images were acquired at the level of papillary muscles. They were then digitally stored for further offline analysis. Left ventricular dimensions were measured using M-mode echocardiography through the short-axis view at the mid-papillary level, and left ventricular ejection fraction (EF) was calculated using the Teicholz method. DAS-RS1echoLAB (Hitachi Healthcare Inc., Tokyo, Japan) was used for the analysis of radial strain (Srad), strain rate (SRrad), and circumferential strain rate (Srcirc). This 2D strain program tracks the movement of speckle patterns due to reflection, scattering, and interference between tissue and ultrasound beams in echocardiographic images. They can be tracked from frame to frame throughout the cardiac cycle. This method allows easy assessment of rotation, torsion, and synchronous disturbance for the left ventricle, as well as strain and strain rate. The endocardial boundary was marked, while the outer boundary was adjusted to fit the epicardial contour. The software (Echolab DAS-RS1) automatically tracked and calculated strain and strain rate in the radial and circumferential directions throughout the six cardiac cycles. Peak systolic Srad, Srrad, and Srcirc were obtained from 6 segments of the papillary muscle levels. Data from at least three different cardiac cycles were averaged. Volume parameters, wall thickness, and internal dimension values were given both as raw data and as body weight ratio index, as previously described [99].

### 4.5. In Vitro Papillary Muscle Experiment

Hearts were excised under terminal 2% isoflurane inhalation anesthesia. Immediately after exsanguination, the hearts were placed in the dissection dish containing heparinized Krebs–Henseleit solution (120 mM NaCl, 4.8 mM KCl, 1.25 mM CaCl_2_·2H_2_O, 1.25 mM MgSO_4_·7H_2_O, 1.2 mM KH_2_PO_4_, 25 mM NaHCO_3_, and 10 mM C_6_H_12_O_6_·H_2_O) gassed with 95% O_2_ and 5% CO_2_ mixture to maintain a pH of 7.4 at 30 °C. Papillary muscles were carefully dissected from the left ventricular wall, avoiding any mechanical tension, and mounted on the force transducer (Commat Pharmacology & Physiology Instruments, Ankara, Türkiye) in the horizontal organ bath chamber. Papillary muscles were perfused continuously with Krebs–Henseleit solution through a peristaltic pump at 5 mL/min. Papillary muscles were stimulated with electric field pulses at 0.6 Hz, 2 ms, twice the threshold voltage value. After a 60-min stabilization period, the papillary muscles were gradually stretched, and L_max_ (maximal contraction value) was recorded. Tension was adjusted to 90% of L_max_ value, and experiments were performed at adjusted tension individually for each preparation. A cumulative concentration–response curve of isoprenaline (0.1 nM–10 µM) was generated. After a washout period, the papillary muscles were challenged with 3 µM forskolin (adenylyl cyclase activator).

### 4.6. In Vitro Langendorff Heart Preparation Experiment

Hearts excised from the rats under terminal 2% isoflurane anesthesia were rapidly cannulated through the aorta and perfused retrogradely with the Krebs–Henseleit solution (120 mM NaCl, 4.8 mM KCl, 1.25 mM CaCl_2_·2H_2_O, 1.25 mM MgSO_4_·7H_2_O, 1.2 mM KH_2_PO_4_, 25 mM NaHCO_3_, and 10 mM C_6_H_12_O_6_·H_2_O). Krebs–Henseleit solution was continuously gassed with a mixture of 95% O_2_ and 5% CO_2_ to maintain a pH of 7.4 at 37 °C. Surrounding adjacent tissue was removed, and the hearts were allowed to beat spontaneously to record the flow rate for each preparation. The hearts were then freed from the atria and paced with an electrode placed in the right ventricle connected to a stimulator (Grass S44 Stimulator, Quincy, MA, USA) to maintain a heart rate of 300 beat/min. An elastic cling film balloon, which was connected to a pressure transducer (Commat Pharmacology & Physiology Instruments, Ankara, Türkiye) via a polyethylene tube, was inserted into the left ventricle to measure left ventricular function. The left ventricular end diastolic pressure (LVEDP) value was set individually for each preparation according to the Frank–Starling curve. Hearts were perfused at constant flow for 30 min to reach a steady state, and a concentration–response curve of CL 316,243, a selective β_3_-AR agonist, (10 pM–1 µM) was generated. Left ventricular function was assessed by the following parameters: left ventricular developed pressure (LVDP), left ventricular end diastolic pressure (LVEDP), rate of contraction (dP/dt_max_), and rate of relaxation (dP/dt_min_).

### 4.7. Western Blot Experiments

Frozen left ventricular tissue was powdered in liquid nitrogen and homogenized with a mixture of RIPA buffer, sodium orthovanadate, and protease inhibitor cocktail. After sonication, samples were agitated at +4 °C for 2 h and centrifuged at 12,000 rpm at +4 °C to obtain the supernatant fraction. The protein concentration of the samples was measured using bicinchoninic acid assay (BCA). Samples containing equal amounts of protein (10 or 45 or 100 µg) were loaded onto 4% acrylamide SDS-PAGE stacking gel and separated by 10% acrylamide SDS-PAGE separation gel. Proteins were transferred to either polyvinylidene difluoride or nitrocellulose membranes at 100 V for 2 h (10 µg), 3 h (45 µg), or 4 h (100 µg). The membranes were blocked with Tris-buffered saline containing 0.1% Tween 20 (TBST) and 5% bovine serum albumin for 1 h at room temperature to prevent non-specific protein binding. The membranes were then incubated with primary antibodies (SERCA2a (1/2000), PLN (1/2000), p-PLN (PLN^ser16/thr17^) (1/1000), or GAPDH (1/5000)) at +4 °C for overnight. The membranes were then washed with TBST and incubated with horseradish peroxidase (HRP)-conjugated antirabbit secondary antibodies (SERCA2a (1/5000), PLN (1/2000), p-PLN (PLN^ser16/thr17^) (1/2000), or GAPDH (1/5000)) at +4 °C for 1.5 h. The membranes were washed with TBST and incubated with enhanced chemiluminescence (ECL) mixture for 1 min, and then the blots were exposed to film. Films were scanned, and protein bands were analyzed by using Image J. GAPDH was used as a housekeeping gene to normalize densitometric values of protein bands; quantitative GAPDH signals in each group are shown in Appendix A.

### 4.8. Data Analysis

Descriptive statistics were performed for each parameter, and data are expressed as mean ± SD. Prism-GraphPad was used for all analyses and generated graphs (version 9.1.2, La Jolla, CA, USA). The efficacy of isoprenaline was determined by fitting a 3-parameter model of the concentration–response curve to obtain estimated E_max_ values. When the same model was applied to derive estimated E_max_ values of CL 316,243, ambiguous values were observed. Therefore, the E_max_ value was considered to be the one obtained at 10 nM CL 316,243, where the highest relaxation was observed. All data were assumed to be normally distributed. Unpaired *t*-test was used to compare C and D groups for model validation. The unpaired *t*-test was used as a hypothesis test to compare D and SV groups and D and V groups for all results obtained in this study. *p*-values < 0.05 were considered statistically significant. Based on the exploratory nature of this study, reported values are to be interpreted as descriptive and not as hypothesis-testing.

Data from two Western blots were pooled and analyzed for GAPDH validation using one-way ANOVA followed by post-hoc Bonferroni test (Appendix A).

### 4.9. Chemicals

HFD and the control diet were purchased from Altromin Spezialfutter GmbH&Co. KG (Lage, Germany) and Arden Research & Experiment (Ankara, Türkiye). Sacubitril/valsartan and valsartan were kindly gifted by Novartis AG (Basel, Switzerland). Isoflurane was purchased from Adeka (Samsun, Türkiye). Chemicals for Krebs–Henseleit solution, STZ, CL 316,243, isoprenaline, and forskolin were from Sigma-Aldrich (Ankara, Türkiye). GAPDH (14C10) primary antibody (CST2118S) (RRID: AB_561053), PLN primary antibody (CST8495) (RRID: AB_10949105), p-PLN (PLN^ser16/thr17^) primary antibody (CST8496) (RRID: AB_10949102), SERCA2a primary antibody (CST4388) (RRID: AB_2227684), and antirabbit secondary antibody (CST7074) (RRID: AB_2099233) were obtained from Cell Signaling (Danvers, Massachusetts, USA). Triglyceride (GPO-PAP) and cholesterol (CHOD-PAP) enzymatic colorimetric kits were purchased from ADS analytic diagnostic systems (Istanbul, Türkiye). The rat insulin ELISA kit (201-11-0708) was purchased from SunRed Biotechnology Company (Shanghai, China).

## 5. Conclusions

Based on the results of our study, the beneficial effects of sacubitril/valsartan and valsartan on EF and FS may be partly due to improved β1- and β2-AR-mediated responses. However, high standard deviations make it difficult to make a definitive statement. Neither treatment reversed diastolic dysfunction or hypertrophy. Given that therapeutic approaches do not improve metabolic parameters, further research is needed to determine whether these drugs may have beneficial effects in the treatment of diabetic cardiac dysfunction.

Limitations of this study are as follows:The age of the animals was considered a baseline characteristic rather than blood glucose levels at the time of randomized group allocation. Therefore, it was not possible to interpret whether sacubitril/valsartan or valsartan affected glycemic control in the present study.Interpretation of the statistical significance data in some experiments was difficult due to large standard deviations. In addition, we were not able to increase the sample size due to ethical constraints and the limited number of animals in the study.Cardiac β-AR subtype mRNA and/or protein expression levels could not be measured due to the non-specific binding capacity of the antibodies [100] and for economic reasons.The possible involvement of components contributing to the β-AR signaling pathways could not be investigated in the present study.

## Figures and Tables

**Figure 1 ijms-25-10617-f001:**
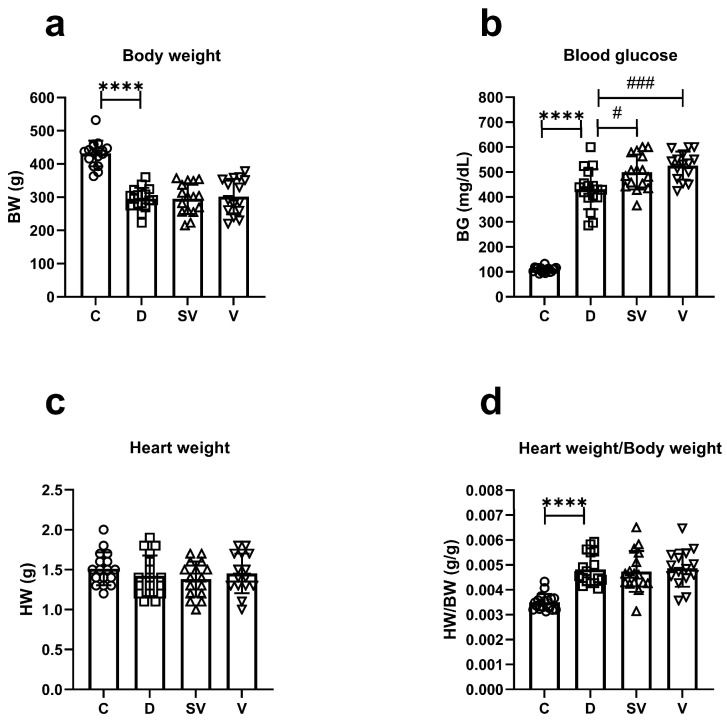
General characteristics at the end of the study. (**a**) Body weight (BW); (**b**) Blood glucose (BG); (**c**) Heart weight (HW); (**d**) The ratio of heart weight to body weight (HW/BW). C, Control (*n* = 17); D, Diabetic (*n* = 17); SV, Sacubitril/valsartan-treated diabetic (*n* = 16); and V, Valsartan-treated diabetic (*n* = 15). ****, *p* < 0.0001 compared to control; #, *p* < 0.05; ###, *p* < 0.001 compared to diabetic.

**Figure 2 ijms-25-10617-f002:**
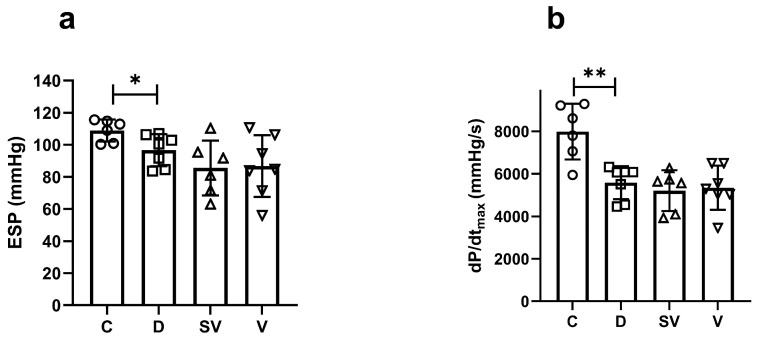
In vivo basal cardiac parameters of systolic function. (**a**) ESP, end systolic pressure; (**b**) dp/dtmax, rate of contraction. C, Control (*n* = 6); D, Diabetic (*n* = 7); SV, Sacubitril/valsartan-treated diabetic (*n* = 6); and V, Valsartan-treated diabetic (*n* = 7). * *p* < 0.05; ** *p* < 0.01 compared to control.

**Figure 3 ijms-25-10617-f003:**
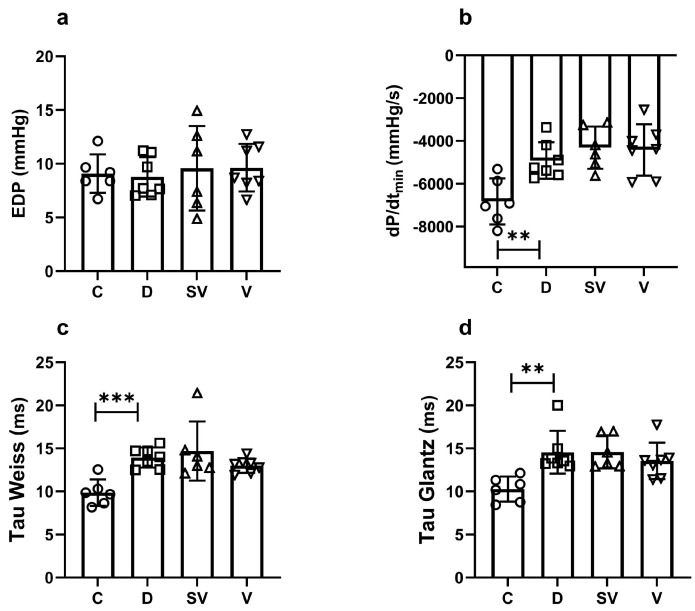
In vivo basal cardiac parameters of diastolic function. (**a**) EDP, end diastolic pressure; (**b**) dP/dtmin, rate of relaxation; (**c**,**d**) Tau, isovolumic relaxation constant. C, Control (*n* = 6); D, Diabetic (*n* = 7); SV, Sacubitril/valsartan-treated diabetic (*n* = 6); and V, Valsartan-treated diabetic (*n* = 7). ** *p* < 0.01; *** *p* < 0.001 compared to control.

**Figure 4 ijms-25-10617-f004:**
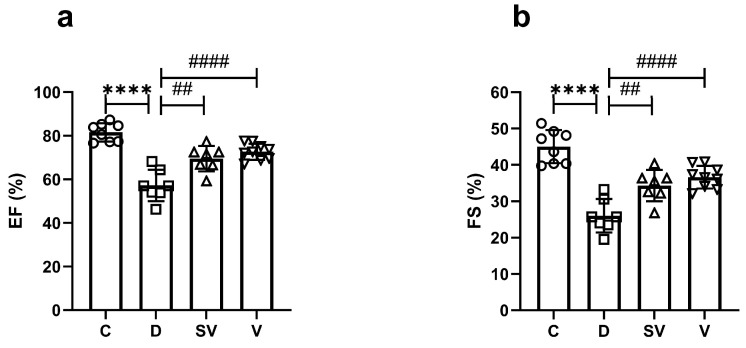
In vivo echocardiography parameters of systolic function. (**a**) % EF, ejection fraction; (**b**) % FS, fractional shortening. C, Control (*n* = 8); D, Diabetic (*n* = 7); SV, Sacubitril/valsartan-treated diabetic (*n* = 7); and V, Valsartan-treated diabetic (*n* = 9). ****, *p* < 0.0001 compared to control. ##, *p* < 0.01; ####, *p* < 0.0001 compared to diabetic.

**Figure 5 ijms-25-10617-f005:**
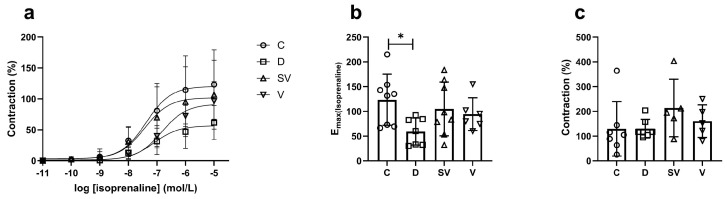
Isoprenaline- and forskolin-mediated contractile responses. (**a**) Cumulative concentration–response curve. (**b**) E_max_, Efficacy. C, Control (*n* = 8); D, Diabetic (*n* = 7); SV, Sacubitril/valsartan-treated diabetic (*n* = 8); and V, Valsartan-treated diabetic (*n* = 6). *, *p* < 0.05 compared to control. (**c**) Contraction response at 3 µM forskolin (% of control). C, Control (*n* = 7); D, Diabetic (*n* = 7); SV, Sacubitril/valsartan-treated diabetic (*n* = 5); and V, Valsartan-treated diabetic (*n* = 5).

**Figure 6 ijms-25-10617-f006:**
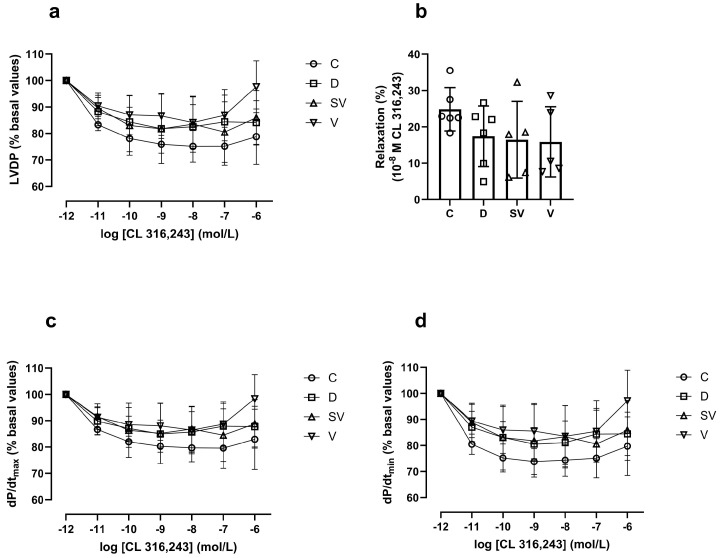
CL 316,243-mediated relaxation responses. (**a**) LVDP, left ventricle developed pressure; (**b**) % relaxation response at 10^−8^ M CL 316,243; (**c**) dP/dt_max_ (% basal values), rate of contraction; (**d**) dP/dt_min_ (%basal values), rate of relaxation. C, Control (*n* = 6); D, Diabetic (*n* = 6); SV, Sacubitril/valsartan-treated diabetic (*n* = 5); and V, Valsartan-treated diabetic (*n* = 5).

**Figure 7 ijms-25-10617-f007:**
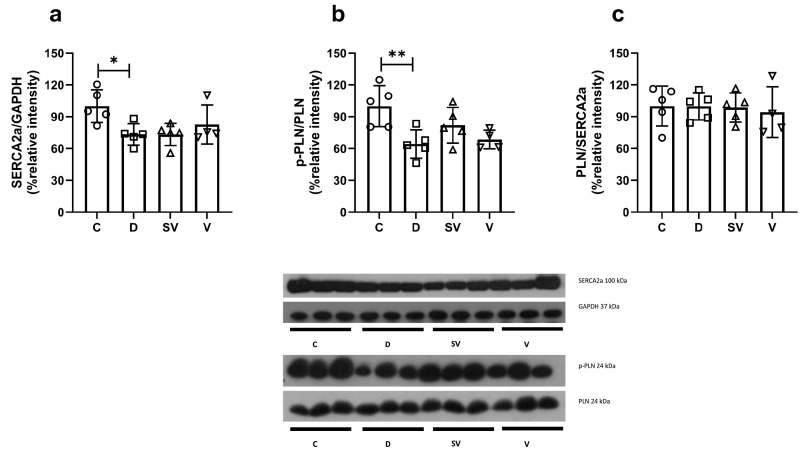
Protein expression levels and representative images. (**a**) SERCA2a protein expression level normalized to GAPDH; (**b**) The ratio of p-PLN to PLN; (**c**) The ratio of PLN to SERCA2a. C, Control (*n* = 5); D, Diabetic (*n* = 5); SV, Sacubitril/valsartan-treated diabetic (*n* = 5); and V, Valsartan-treated diabetic (*n* = 4). *, *p* < 0.05; **, *p* < 0.01 compared to control.

**Table 1 ijms-25-10617-t001:** In vivo basal cardiac hemodynamic parameters.

	C (*n* = 6)	D (*n* = 7)	SV (*n* = 6)	V (*n* = 7)
SBP (mmHg)	110.89 ± 18.41	103.40 ± 11.69	93.54 ± 19.72	91.52 ± 20.41
DBP (mmHg)	79.29 ± 22.43	71.94 ± 9.23	67.69 ± 16.18	65.99 ± 20.57
MAP (mmHg)	100.36 ± 19.55	92.91 ± 10.77	84.92 ± 18.24	83.01 ± 20.13
HR (beat/min)	311.13 ± 39.09	247.17 ± 20.27 **	256.29 ± 18.74	251.35 ± 21.76
EDV (µL)	370.64 ± 69.67	436.66 ± 46.89	436.09 ± 76.22	411.52 ± 74.01
EDVI (µL/g)	0.92 ± 0.16	1.56 ± 0.29 ***	1.59 ± 0.31	1.43 ± 0.29
ESV (µL)	162.06 ± 37.83	200.17 ± 26.07	209.34 ± 26.93	191.61 ± 22.42
ESVI (µL/g)	0.40 ± 0.08	0.71 ± 0.13 ***	0.77 ± 0.18	0.67 ± 0.11
SV (µL)	208.58 ± 47.93	236.49 ± 36.31	226.75 ± 60.39	219.91 ± 60.81
SVI (µL/g)	0.52 ± 0.12	0.85 ± 0.19 **	0.82 ± 0.18	0.76 ± 0.21

SBP, systolic blood pressure; DBP, diastolic blood pressure; MAP, mean arterial pressure; HR, heart rate; EDV, end diastolic volume; EDVI, end diastolic volume index; ESV, end systolic volume; ESVI, end systolic volume index; SV, stroke volume; SVI, stroke volume index; C, Control (*n* = 6); D, Diabetic (*n* = 7); SV, Sacubitril/valsartan-treated diabetic (*n* = 6); and V, Valsartan-treated diabetic (*n* = 7). **, *p* < 0.01; ***, *p* < 0.001, compared to control.

**Table 2 ijms-25-10617-t002:** Preload independent in vivo cardiac parameters.

	C (*n* = 5)	D (*n* = 6)	SV (*n* = 6)	V (*n* = 6)
ESPVR	0.390 ± 0.156	0.449 ± 0.166	0.334 ± 0.106	0.317 ± 0.132
EDPVR	0.010 ± 0.002	0.007 ± 0.003	0.006 ± 0.003	0.006 ± 0.003
PRSW	62.58 ± 9.44	53.69 ± 6.83	56.86 ± 7.73	53.31 ± 3.79

ESPVR, end systolic pressure–volume relationship; EDPVR, end diastolic pressure–volume relationship; PRSW, preload recruitable stroke work; C, Control (*n* = 5); D, Diabetic (*n* = 6); SV, Sacubitril/valsartan-treated diabetic (*n* = 6); and V, Valsartan-treated diabetic (*n* = 6).

**Table 3 ijms-25-10617-t003:** In vivo echocardiography parameters.

	C (*n* = 8)	D (*n* = 7)	SV (*n* = 7)	V (*n* = 9)
IVSd (mm)	1.91 ± 0.58	2.31 ± 0.38	1.83 ± 0.25 #	1.78 ± 0.25 ##
IVSId (mm/kg)	4.30 ± 1.37	6.94 ± 1.04 **	5.64 ± 0.65 #	5.73 ± 1.13 #
LVIDd (mm)	5.49 ± 0.67	5.06 ± 0.91	5.21 ± 1.12	4.92 ± 0.50
LVIDId (mm/kg)	12.26 ± 1.55	15.33 ± 3.35 *	16.46 ± 5.39	15.84 ± 2.52
LVPWd (mm)	2.35 ± 1.90	2.20 ± 0.72	1.89 ± 0.29	2.01 ± 0.66
LVPWId (mm/kg)	5.05 ± 3.47	6.51 ± 1.65	5.82 ± 0.87	6.47 ± 2.30
IVSs (mm)	2.93 ± 0.83	2.76 ± 0.51	2.17 ± 0.45 #	2.17 ± 0.35 #
IVSIs (mm/kg)	6.50 ± 1.69	8.31 ± 1.65	6.76 ± 1.65	6.96 ± 1.37
LVIDs (mm)	3.01 ± 0.21	3.71 ± 0.51 **	3.43 ± 0.78	3.10 ± 0.21 ##
LVIDIs (mm/kg)	6.75 ± 0.72	11.27 ± 2.20 ****	10.86 ± 3.80	9.96 ± 1.16
LVPWs (mm)	2.93 ± 0.49	2.80 ± 0.66	2.56 ± 0.54	2.79 ± 0.60
LVPWIs (mm/kg)	6.57 ± 1.36	8.33 ± 1.43 *	7.80 ± 1.09	9.00 ± 2.29
CO (mL/dk)	101.25 ± 39.07	52.86 ± 29.84 *	60.14 ± 28.64	51.11 ± 13.64
CI (mL/dk.g)	0.23 ± 0.09	0.16 ± 0.09	0.19 ± 0.11	0.16 ± 0.05
EF (%)	81.60 ± 4.20	57.27 ± 7.21 ****	69.56 ± 5.84 ##	72.69 ± 3.72 ####
FS (%)	45.01 ± 4.54	26.04 ± 4.60 ****	34.34 ± 4.31 ##	36.63 ± 3.11 ####

IVSd, interventricular septum thickness at end diastole; IVSId, interventricular septum thickness at end diastole index; LVIDd, left ventricular internal dimension at end diastole; LVIDId, left ventricular internal dimension at end diastole index; LVPWd, left ventricular posterior wall thickness at end diastole; LVPWd, left ventricular posterior wall thickness at end diastole index; IVSs, interventricular septum thickness at end systole; IVSIs, interventricular septum thickness at end systole index; LVIDs, left ventricular internal dimension at end systole; LVIDIs, left ventricular internal dimension at end systole index; LVPWs, left ventricular posterior wall thickness at end systole; LVPWIs, left ventricular posterior wall thickness at end systole index; CO, cardiac output; CI, cardiac index; % EF, ejection fraction; % FS, fractional shortening; C, Control (*n* = 8); D, Diabetic (*n* = 7); SV, Sacubitril/valsartan-treated diabetic (*n* = 7); and V, Valsartan-treated diabetic (*n* = 9). *, *p* < 0.05; **, *p* < 0.01; ****, *p* < 0.0001 compared to control. #, *p* < 0.05; ##, *p* < 0.01; ####, *p* < 0.001 compared to diabetic.

## Data Availability

The raw data supporting the conclusions of this article will be made available by the authors on request.

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
