# Peer review of "Sacubitril/Valsartan Combination Partially Improves Cardiac Systolic, but Not Diastolic, Function through β-AR Responsiveness in a Rat Model of Type 2 Diabetes"

_ijms, 2024, doi:10.3390/ijms251910617_

Round 1

Reviewer 1 Report

Comments and Suggestions for Authors

The manuscript presents interesting results; however, some clearer specifications should be made.

1.     in Abstract (lines 1-2) you mentioned “Cardiovascular complications are the major cause of  diabetes-related morbidity and mortality.

From a certain point of view, what you say is ok, but somehow you omit to specify the role played by endothelial dysfunction - the major link between diabetes and CV diseases. Neither the abstract nor the article shows this. Without endothelial dysfunction due to hyperglycemia, it is more difficult to be affected the heart or vessels.

Please consider this suggestion and insert it in the article where you think it is most appropriate.

2.     In the abstract an in the article (discussion, page 11 etc.) you mentioned “…relaxation response…” without mentioning more clearly what kind of relaxation you were referring to (whose relaxation is it). There is also relaxation of the vascular smooth muscles.

3.     I think that in the article you are referring to diabetes and not to another form of diabetes. Please mention this at least after the first use of the word diabetes.

 4.     In introduction you mentioned “Decreased levels of NPs are associated with obesity [13], insulin resistance [14] and the development of diabetes [15]. Therefore, it was proposed that therapeutic agents that increase NPs may be beneficial in the treatment of diabetes and its complications”.

It should be mentioned very briefly why inhibition of NPs degradation (inhibition of neprilizyn) is beneficial in the treatment of DM or as stated below in the introduction to improve glucose homeostasis.

You must also refer a little to the mechanisms of action of the substances, not only to the effects/results obtained or presented in the literature.

5.     The exposure/presentation of receptors β, made in the introduction, is not very clear. Cardiac activity is mainly influenced by β1. The β2 receptor is insignificant on cardiac functions, and the β3 receptor is less relevant for your context.

And if the expression of β1 and β2 receptors is reduced in DM, why is the administration of beta-blockers contraindicated in diabetics?

The presentation you made of beta receptors in the context of DM and some cardiac problems-related to DM is not very clear and relevant for your experimental study.

This part should be improved

6.     Even if you refer to a possible connection between sacubitril/valsartan and β receptors, this connection is not clear highlighted either in the introduction or in the results obtained or discussion. This is an aspect that must be improved.

7.     Everywhere in the article (materials and methods, results, discussions) you mention plasma glucose level without mentioning anywhere that it is synonymous with glycemia.

8.     To my knowledge, the OGTT does not involve insulin dosing. You determined the OGTT and the IR assessment (HOMA-IR index). Perhaps it should be mentioned that there are 2 different tests/analyses that complement each other, it is not just the OGTT - page 12.

The same thing should be clarified in the discussions.

9.     It is recommended to mention in the text what the abbreviations used represent (page 13), not only in tables and results section.

10.  Are you sure that you determined dP/dt max and dP/dt min using the Langhendorff technique? – page 13

These parameters are rather determined by cardiac catheterization.

11.  SV and V groups were not compared with C but only with D? If you used Anova one-way to interpret the results from the Wester blot, why didn't you use the Anova test for the other results as well? Thus, you could compare all 4 groups with each other. – page 14

12.  Maybe the limitations of the study would fit at the end of the discussions if the journal has no other specific requirements.

13.  I think that the statistical analysis for figure 1 b (blood glucose; SV vs D; V vs D and V vs SV) is not correctly done or interpreted – page 3.

In this sense, you can compare the statistical interpretation from graph 1b with that from 1d. Please check this aspect.

14.  At page 6 where you explain evolution of EF (%), you affirmed that EF is markedly reduced in diabetic rats indicating/highlighting an impairment of systolic function and its improvement in SV and S groups by the treatment used.

Yes, you obtained these results, but according to the European HF guideline, we have systolic dysfunction (HFrEF) or a significant impairment of systolic function starting with a decrease in EF below 40% (even in experimental studies).

So you obtained some changes in the EF in the diabetic rats respectively under the applied treatment, but you should mentioned that the EF values ​​did not drop below 50-60%, so you do not have an important systolic function alteration or systolic dysfunction.

It is not recommended that readers understand that in diabetic animals FE (%) decreased as much as in HF and the treatment used significantly increased/attenuated this.

The same idea is found in the discussions on page 10 (e.g. systolic dysfunction). Please reword taking into account the provisions of the HF guidelines.

15.  You could enlarge the graphs in figures 5, 6 and 7 to make the pictures clearer.

16.  Discussion: diastolic function and diastolic dysfunction is characterized by different parameters such as E/A ratio, e'/a' ratio, E/e' ratio etc. From what I understood, you did not determine them. When you state an aspect related to diastolic function or diastolic dysfunction and compare it with other results obtained in other studies, perhaps it would be correct to briefly recall how you characterized it and how other studies characterized this diastolic function (discussion page 10). Otherwise we compare different things.

You say (discussion page 10) that some differences between yours results and those form bibliography come from the differences in the animal models. They can also come from the parameters used to highlight the respective dysfunction (e.g. diastolic dysfunction).

17.  In discussion you say: “Sacubitril/valsartan and valsartan were comparable and slightly increased ß1- and ß2-AR mediated contractile responses, which were impaired in diabetic rat”.

It would be good to detail more clearly, how this was observed in your study. Although this aspect is mentioned and emphasized from the title of the manuscript, it is not clear to me how you reached this conclusion (involvement of β1 and β2 receptors in the contractile response of SV or V). At the moment, I consider this statement a bit bold and unjustified.

The same idea is found in the discussions, on page 11.

I do not consider that the explanations currently proposed by you are sufficient.

If you want the article to remain in this note, you will have to explain more clearly the possible connection between SV and V with β-AR receptors. If not, you will have to reformulate the title.

18.  Do you think that the glycemia increased in the SV and V groups due to the stress produced by the gavage? Here you know best what the working conditions were, but due to the stress induced by the gavage for 4 weeks, it is possible that glycemia increased. Were groups C and D also gavaje with the solvent? the manuscript does not mention this.

19.  When you talk in discussions about hypertrophy, why don't you also mention the thickness of the interventricular septum and the thickness of ventricular wall? I think it is more relevant than the heart weigh /body weight ratio.

Perhaps it should be mentioned by which mechanism it is considered that SV or S could intervene to prevent or reduce cardiac hypertrophy.

20.  Page 11: Ca2+ influx into cardiomyocytes induces cellular calcium release from which level? Where does this intracellular calcium come from? Maybe it should be specified in the article.

21.  In the discussions you compared the results obtained by you with other published results, which is a very good aspect, but I suggest you also offer some possible explanations of the results obtained starting from the action mechanism of the 2 compounds.

22.  The conclusions (page 15) should be formulated more clearly and concisely according to the results obtained in all 4 groups and in accordance with the changes made in the manuscript.

Author Response

Comments 1: In Abstract (lines 1-2) you mentioned “Cardiovascular complications are the major cause of diabetes-related morbidity and mortality.”

From a certain point of view, what you say is ok, but somehow you omit to specify the role played by endothelial dysfunction- the major link between diabetes and CV diseases. Neither the abstract nor the article shows this. Without endothelial dysfunction due to hyperglycemia, it is more difficult to be affected the heart or vessels.

Please consider this suggestion and insert it in the article where you think it is most appropriate.

Response 1: Relevant information was added in introduction as suggested by the reviewer (line 60-76)

Comments 2: In the abstract an in the article (discussion, page 11 etc.) you mentioned “…relaxation response…” without mentioning more clearly what kind of relaxation you were referring to (whose relaxation is it). There is also relaxation of the vascular smooth muscles.

Response 2: It was reworded as "cardiac relaxation". (line 35, 423, 424, 426 and 433).

Comments 3: I think that in the article you are referring to diabetes and not to another form of diabetes. Please mention this at least after the first use of the word diabetes.

Response 3: Upon first mentioning, diabetes was reworded to diabetes mellitus (see line 18 and 53).

Comments 4: In introduction you mentioned “Decreased levels of NPs are associated with obesity [13], insulin resistance [14] and the development of diabetes [15]. Therefore, it was proposed that therapeutic agents that increase NPs may be beneficial in the treatment of diabetes and its complications”.

It should be mentioned very briefly why inhibition of NPs degradation (inhibition of neprilizyn) is beneficial in the treatment of DM or as stated below in the introduction to improve glucose homeostasis.

You must also refer a little to the mechanisms of action of the substances, not only to the effects/results obtained or presented in the literature.

Response 4 : Relevant information was added in introduction as suggested by the reviewer (line 85-91).

Comments 5: The exposure/presentation of receptors β, made in the introduction, is not very clear. Cardiac activity is mainly influenced by β1. The β2 receptor is insignificant on cardiac functions, and the β3 receptor is less relevant for your context.

And if the expression of β1 and β2 receptors is reduced in DM, why is the administration of beta-blockers contraindicated in diabetics?

The presentation you made of beta receptors in the context of DM and some cardiac problems-related to DM is not very clear and relevant for your experimental study.

This part should be improved

Response 5 : The only pro-inotropic subtype in rats is β1; however, β2 also play a role in the human heart as demonstrated in vitro and in vivo by several groups. See e.g., Brodde & Michel 1999 Pharmacol Rev 51: 651-689. β3-AR indeed only are relevant for relaxation (Arioglu-Inan 2019 Br J Pharmacol 176: 2482-2495 and Erdogan 2020 Cells 9: 2548). However, the overall function of the diabetic heart depend on the balance between positive and negative inotropic stimuli.

The contraindication of β-blockers in diabetes is relative, not absolute. It relates to the potential of β-blockers to worsen the metabolic situation.

We did not propose a specific role of β-AR. Rather these generally are the most important mediators of positive inotropic effects; therefore, exploring possible effects of treatment on β-AR is relevant.

Comments 6: Even if you refer to a possible connection between sacubitril/valsartan and β receptors, this connection is not clear highlighted either in the introduction or in the results obtained or discussion. This is an aspect that must be improved.

Response 6 : There is no direct connection between sacubitril/valsartan and cardiac β-AR. However, because of impairment of cardiac β-AR in many diabetes models, it was relevant to ask whether the beneficial effect of treatment involves improved β-AR function. It was implemented in introduction and discussion (line 112-114 and line 284-285).

Comments 7: Everywhere in the article (materials and methods, results, discussions) you mention plasma glucose level without mentioning anywhere that it is synonymous with glycemia.

Response 7 : Glycemia is a qualitative term in this regard. Plasma glucose levels are a quantitative indicator of the glycemic situation. It was implemented in the text (please see 4.1. Animals and the study protocol section, line 482).

Comments 8: To my knowledge, the OGTT does not involve insulin dosing. You determined the OGTT and the IR assessment (HOMA-IR index). Perhaps it should be mentioned that there are 2 different tests/analyses that complement each other, it is not just the OGTT - page 12.

The same thing should be clarified in the discussions.

Response 8 : Of course, OGTT does not involve insulin administration. Rather it measures the ability of the body to react to an acute glucose load. This is considered a more sensitive indicator of dysfunction than glucose levels alone.

Comments 9: It is recommended to mention in the text what the abbreviations used represent (page 13), not only in tables and results section.

Response 9 : The journal's guideline for authors suggest that abbreviations should be defined the first time they appear in each of three sections: the abstract, the main text, and the first figure or table. However, for the convenience of the reader, we have redefined the abbreviations as recommended by the reviewer (line 500, 508-511, 513-515, 528, 575-577).

Comments 10: Are you sure that you determined dP/dt max and dP/dt min using the Langhendorff technique? – page 13 These parameters are rather determined by cardiac catheterization.

Response 10 : The Langendorff heart preparation experiment is a well-known method for determining these parameters in preclinical animal studies.  As an example, here are some papers that use this technique to determine dP/dtmax and dP/dtmin.

  1. Gu, X., Xu, J., Zhu, L., Bryson, T., Yang, X. P., Peterson, E., & Harding, P. (2016). Prostaglandin E2 reduces cardiac contractility via EP3 receptor. Circulation: Heart Failure, 9(8), e003291.
  2. Singh, L., Virdi, J. K., Maslov, L. N., Singh, N., & Jaggi, A. S. (2018). Investigating the possible mechanisms involved in adenosine preconditioning‐induced cardioprotection in rats. Cardiovascular Therapeutics, 36(3), e12328.
  3. Güven, B., Kara, Z., & Onay‐Beşikci, A. (2020). Metabolic effects of carvedilol through β‐arrestin proteins: investigations in a streptozotocin‐induced diabetes rat model and in C2C12 myoblasts. British Journal of Pharmacology, 177(24), 5580-5594.
  4. Salameh, A., Keller, M., Dähnert, I., & Dhein, S. (2017). Olesoxime inhibits cardioplegia-induced ischemia/reperfusion injury. A study in langendorff-perfused rabbit hearts. Frontiers in physiology, 8, 324.

Comments 11: SV and V groups were not compared with C but only with D? If you used Anova one-way to interpret the results from the Western blot, why didn't you use the Anova test for the other results as well? Thus, you could compare all 4 groups with each other. – page 14

Response 11 : As mentioned in the data analysis section, we analysed all results using an unpaired t-test. The reason for this is that the more comparisons are made, the lower the statistical power becomes. The comparison of D vs. C establishes whether the disease causes a change; SV and V are only compared to D because the question here is whether they improve the situation in D. There was no comparison of SV or V with C because the that was not a question of the study and would have lowered the statistical power.

One-way ANOVA followed by Bonferroni post-tests was only used to demonstrate that there was comparable GAPDH protein expression between groups to ensure that it was a suitable housekeeping gene in our study.

Comments 12: Maybe the limitations of the study would fit at the end of the discussions if the journal has no other specific requirements.

Response 12 : We prefer to mention the limitations at the beginning of the Discussion because it allows readers to follow our interpretations while being fully aware of possible limitations.

Comments 13: I think that the statistical analysis for figure 1 b (blood glucose; SV vs D; V vs D and V vs SV) is not correctly done or interpreted – page 3.

In this sense, you can compare the statistical interpretation from graph 1b with that from 1d. Please check this aspect.

Response 13 : It did not fully become clear to us what the reviewer means by this comment. We applied the same statistical approach for all parameters. This approach had been pre-specified by one of the authors (MCM) who is the professor in charge of teaching statistics to PhD students in the life sciences both in Mainz and Ankara; moreover, he is the statistics editor of a major pharmacology journal, Mol Pharmacol.

Comments 14: At page 6 where you explain evolution of EF (%), you affirmed that EF is markedly reduced in diabetic rats indicating/highlighting an impairment of systolic function and its improvement in SV and S groups by the treatment used.

Yes, you obtained these results, but according to the European HF guideline, we have systolic dysfunction (HFrEF) or a significant impairment of systolic function starting with a decrease in EF below 40% (even in experimental studies).

So you obtained some changes in the EF in the diabetic rats respectively under the applied treatment, but you should mentioned that the EF values did not drop below 50-60%, so you do not have an important systolic function alteration or systolic dysfunction.

It is not recommended that readers understand that in diabetic animals FE (%) decreased as much as in HF and the treatment used significantly increased/attenuated this.

The same idea is found in the discussions on page 10 (e.g. systolic dysfunction). Please reword taking into account the provisions of the HF guidelines.

Response 14 : We do not feel that the EF-based definitions of types of HF for humans can directly be applied to rats. For instance, among the many reasons is that rat heart operates at a much higher frequency (about 300 bpm or more) than the human heart. Therefore, the guideline mentioned is not applicable to our study.

Comments 15: You could enlarge the graphs in figures 5, 6 and 7 to make the pictures clearer.

Response 15 : The size of the graphs is defined by the template the journal asks to use. We have used the maximum size allowed by this template.

Comments 16: Discussion: diastolic function and diastolic dysfunction is characterized by different parameters such as E/A ratio, e'/a' ratio, E/e' ratio etc. From what I understood, you did not determine them. When you state an aspect related to diastolic function or diastolic dysfunction and compare it with other results obtained in other studies, perhaps it would be correct to briefly recall how you characterized it and how other studies characterized this diastolic function (discussion page 10). Otherwise we compare different things.

You say (discussion page 10) that some differences between yours results and those form bibliography come from the differences in the animal models. They can also come from the parameters used to highlight the respective dysfunction (e.g. diastolic dysfunction).

Response 16 : It is true that we could not measure the E/A, e/e' ratios, which are markers of diastolic function by echocardiography, due to the inadequacy of our equipment. However, the occurrence of diastolic dysfunction in the models and the effect of the treatment approach on diastolic function can be interpreted independently of the method used. We do not believe that the difference between the methods affects the comparability of the results. However, as suggested by the reviewer, we have made the necessary additions to the text to make it clearer for the reader (line 385-388).

Comments 17: In discussion you say: “Sacubitril/valsartan and valsartan were comparable and slightly increased ß1- and ß2-AR mediated contractile responses, which were impaired in diabetic rat”.

It would be good to detail more clearly, how this was observed in your study. Although this aspect is mentioned and emphasized from the title of the manuscript, it is not clear to me how you reached this conclusion (involvement of β1 and β2 receptors in the contractile response of SV or V). At the moment, I consider this statement a bit bold and unjustified.

The same idea is found in the discussions, on page 11.

I do not consider that the explanations currently proposed by you are sufficient.

If you want the article to remain in this note, you will have to explain more clearly the possible connection between SV and V with β-AR receptors. If not, you will have to reformulate the title.

Response 17 : Isoprenaline is known to elicit contractile responses in the heart via β1-Ars (and in humans β2-Ars). Papillary muscle is known as a beneficial tool to study cardiac contractility (Uhl et al., 2015). Moreover, isoprenaline induced positive inotropic effect in papillary muscle has been used to comment on β-AR mediated contractile response (Ferrara et al., 2005; Amour et al., 2007; Carillion et al., 2017). We conclude that the beneficial effects of sacubitril/valsartan treatment on systolic function (as it refers to cardiac contractility) may be related to slightly improved β-AR-mediated contractile responses, based on cardiac haemodynamic parameters and contractile responses obtained from papillary muscle experiments (line 288, 390-391). However, as mentioned in the limitations section, it was not possible to support these data with molecular experiments (density of β-AR subtypes). The title has been changed to exclude β-AR subtypes. We believe that this title comprehensively reflects the results of our study as it now stands (line 3).

Sebastian Uhl,  Marc Freichel,  and Ilka Mathar, Contractility Measurements on Isolated Papillary Muscles for the Investigation of Cardiac Inotropy in Mice. J Vis Exp. 2015; (103): 53076.

Nicola Ferrara, Pasquale Abete, Graziamaria Corbi, Giuseppe Paolisso, Giancarlo Longobardi, Claudio Calabrese, Francesco Cacciatore, Donatella Scarpa, Guido Iaccarino, Bruno Trimarco,

Dario Leosco, Franco Rengo. Insulin-induced changes in β-adrenergic response: An experimental study in the isolated rat papillary muscle. American Journal of Hypertension, Volume 18, Issue 3, March 2005, Pages 348–353.

Amour, J.; Loyer, X.; Le Guen, M.; Mabrouk, N.; David, J.S.; Camors, E.; Carusio, N.; Vivien, B.; Andriantsitohaina, R.; Heymes, C.; et al. Altered contractile response due to increased beta3-adrenoceptor stimulation in diabetic cardiomyopathy: The role of nitric oxide synthase 1-derived nitric oxide. Anesthesiology 2007, 107, 452–460.

Carillion, A.; Feldman, S.; Na, N.; Biais, M.; Carpentier, W.; Birenbaum, A.; Cagnard, N.; Loyer, X.; Bonnefont-Rousselot, D.; Hatem, S.; et al. Atorvastatin reduces beta-Adrenergic dysfunction in rats with diabetic cardiomyopathy. PLoS ONE 2017, 12, e0180103.

Comments 18: Do you think that the glycemia increased in the SV and V groups due to the stress produced by the gavage? Here you know best what the working conditions were, but due to the stress induced by the gavage for 4 weeks, it is possible that glycemia increased. Were groups C and D also gavaje with the solvent? the manuscript does not mention this.

Response 18 : We can exclude the option of a gavage-induced stress because the other groups also received gavage, just with vehicle. It was implemented in the text (please see 4.1. Animals and the study protocol section, line 479).

Comments 19: When you talk in discussions about hypertrophy, why don't you also mention the thickness of the interventricular septum and the thickness of ventricular wall? I think it is more relevant than the heart weight /body weight ratio.

Perhaps it should be mentioned by which mechanism it is considered that SV or S could intervene to prevent or reduce cardiac hypertrophy.

Response 19 : Rat studies typically assess cardiac hypertrophy by organ weight (Gunadi et al., 2019; Sukumaran et al., 2017, Xiang et al., 2005, Zapata-Sudo et al., 2014).  This is contrast to humans where only non-invasive measures such as echo can be used for which wall thickness is more appropriate.

Gunadı, J. W., Tarawan, V. M., Setıawan, I., Lesmana, R., Wahyudıanıngsıh, R. & Supratman, U. 2019. Cardiac hypertrophy is stimulated by altered training intensity and correlates with autophagy modulation in male Wistar rats. BMC Sports Sci Med Rehabil, 11, 9.

Sukumaran, A., Chang, J., Han, M., Mıntrı, S., Khaw, B. A. & Kım, J. 2017. Iron overload exacerbates age-associated cardiac hypertrophy in a mouse model of hemochromatosis. Sci Rep, 7, 5756.

Xıang, W., Kong, J., Chen, S., Cao, L. P., Qıao, G., Zheng, W., Lıu, W., Lı, X., Gardner, D. G. & Lı, Y. C. 2005. Cardiac hypertrophy in vitamin D receptor knockout mice: role of the systemic and cardiac renin-angiotensin systems. Am J Physiol Endocrinol Metab, 288, E125-32.

Zapata-Sudo, G., Da Sılva, J. S., Pereıra, S. L., Souza, P. J., De Moura, R. S. & Sudo, R. T. 2014. Oral treatment with Euterpe oleracea Mart. (acai) extract improves cardiac dysfunction and exercise intolerance in rats subjected to myocardial infarction. BMC Complement Altern Med, 14, 227.

Comments 20: Page 11: Ca2+ influx into cardiomyocytes induces cellular calcium release from which level? Where does this intracellular calcium come from? Maybe it should be specified in the article.

Response 20 : Relevant information was added in discussion as suggested by the reviewer (line 435-437)

Comments 21: In the discussions you compared the results obtained by you with other published results, which is a very good aspect, but I suggest you also offer some possible explanations of the results obtained starting from the action mechanism of the 2 compounds.

Response 21 : We've included possible explanations for the differences between our study and other studies, where applicable (line 337-342, 350-358, 399-417).

Comments 22: The conclusions (page 15) should be formulated more clearly and concisely according to the results obtained in all 4 groups and in accordance with the changes made in the manuscript.

Response 22 : We've changed the conclusion as suggested by the reviewer. However, no changes have been made to the manuscript to alter the conclusion, as can be seen from our responses to the reviewer's comments (see R1.11).

Reviewer 2 Report

Comments and Suggestions for Authors

The article “Sacubitril/valsartan combination partially improves cardiac systolic function through β1-/β2-AR responsiveness but not diastolic function in a rat model of type 2 diabetes”, is related to the effect of inhibition of the renin-angiotensin system and neprilysin, in which it is observed that the β1-/β2 receptors participate, the article is adequate, however, it is necessary to improve the following aspects:

In the introduction, it is necessary to clarify the mechanisms involved that lead to studying the effect of sacubitril and valsartan.

The results are adequate and presented correctly

However, the discussion is very general, although it is based on references, there is no proposed mechanism.

It is necessary to clarify why there is a decrease in weight, the effect of glucose, as well as cholesterol and triglyceride levels because there is no mechanism explanation.

Changes in beta adrenergic responses require postulating a mechanism, not just mentioning calcium and sympathoinhibition.

The conclusion seems like a summary of results. it is necessary to modify it

Author Response

Comments 1: In the introduction, it is necessary to clarify the mechanisms involved that lead to studying the effect of sacubitril and valsartan.

Response 1: Sacubitril and valsartan are guideline recommended HF treatments. The purpose of our study was not to demonstrate whether they work. Rather we focused on whether they affect  β-AR function in the heart.

Comments 2: The results are adequate and presented correctly.  However, the discussion is very general, although it is based on references, there is no proposed mechanism.

Response 2: We explored whether the two treatments may perhaps work by improving β-AR function in the heart. We provide data on some mechanisms such as intracellular free Ca2+ levels. It was not the intention of our study to determine the molecular link between the two treatments and the observed effects. To study those, it was first necessary to explore whether improvements at the level of β-AR function occur. Investigating the cellular and molecular mechanism of such improvements would required a distinct additional study.

Comments 3: It is necessary to clarify why there is a decrease in weight, the effect of glucose, as well as cholesterol and triglyceride levels because there is no mechanism explanation.

Response 3: The observed changes in body weight and the metabolic parameters are established over dozens of studies for the STZ model of diabetes, which is based on destruction of pancreatic beta cells; we show them here only as model validation.

Comments 4: Changes in beta adrenergic responses require postulating a mechanism, not just mentioning calcium and sympathoinhibition.

Response 4: As stated above, we respectfully disagree. Firstly, changes in  β-AR function have already been studied repeatedly (see e.g., a review Erdogan 2020 Cells 9: 2548). Second, our study was designed to address the question whether known beneficial effects of sacubitril and valsartan involved improvements of β-AR function. How the two treatments do that is a distinct question that will require another type of study.

Comments 5: The conclusion seems like a summary of results. it is necessary to modify it

Response 5: It was modified as suggested by the reviewer (line 629-635).

Round 2

Reviewer 1 Report

Comments and Suggestions for Authors

Dear authors,

Certain questions or suggestions were addressed to the authors in order to improve the quality of the manuscript.

I considered that things should be clarified or expressed a bit more clearly, first of all in order not to mislead the readers.

What each of us, as authors and reviewers, think or feel is relevant in this context.

If there are clear guidelines that define the characterization/diagnostic criteria of a disease, it is completely irrelevant what some of us think/feel - speaking of the answer given on EF (%).

There are reference articles performed on laboratory animals (rats, mice) that characterize HF according to values ​​or reference intervals of EF (%) from humans.

I have left you only 2 bibliographic indexes below - but there are a lot of publications in this sense.

1.     Dhot, J. et al. Overexpression of endothelial β3‐adrenergic receptor induces diastolic dysfunction in rats. ESC Heart Failure 2020; 7: 4159–4171; DOI: 10.1002/ehf2.13040

2.     Schiattarella, G.G.; Altamirano, F.; Tong, D.; French, K.M.; Villalobos, E.; Kim, S.Y.; Luo, X.; Jiang, N.; May, H.I.; Wang, Z.V.; et al. Nitrosative stress drives heart failure with preserved ejection fraction. Nature 2019, 568, 351–356; doi.org/10.1038/s41586-019-1100-z

In your manuscript you cannot say that you have systolic dysfunction, it is an error.

You registered a reduction in EF (%) but you are not yet on systolic dysfunction!

In the same articles recommended, you can also find the characterization of diastolic dysfunction and cardiac hypertrophy

Also, some small recommendations for a better/clear form of the manuscript:

1.     The introduction is too long, the information should be compressed somehow or you should drop irrelevant aspects.

2.     What do you understand by cardiac relaxation? do you mean diastole or what exactly are you referring to? Don't specify that anywhere.

3.     line 64: usually, hyperglycemia increases the blood volume (glucose increases osmotic pressure and attracts water)

4.     lines 33-68: Cardiac metabolism is based on fatty acid oxidation (~80%), so it is not a big problem for the heart if it does not have access to glucose under basal conditions. Under conditions of stress such as ischemia, the situation may be different.
